

# Identification and temporal expression of putative circadian clock transcripts in the amphipod crustacean *Talitrus saltator*

Joseph F. O'Grady, Laura S. Hoelters, Martin T. Swain and David C. Wilcockson

Institute of Biological, Environmental and Rural Sciences, University of Wales, Aberystwyth, Ceredigion, United Kingdom

## ABSTRACT

**Background**. *Talitrus saltator* is an amphipod crustacean that inhabits the supralittoral zone on sandy beaches in the Northeast Atlantic and Mediterranean. *T. saltator* exhibits endogenous locomotor activity rhythms and time-compensated sun and moon orientation, both of which necessitate at least one chronometric mechanism. Whilst their behaviour is well studied, currently there are no descriptions of the underlying molecular components of a biological clock in this animal, and very few in other crustacean species.

**Methods**. We harvested brain tissue from animals expressing robust circadian activity rhythms and used homology cloning and Illumina RNAseq approaches to sequence and identify the core circadian clock and clock-related genes in these samples. We assessed the temporal expression of these genes in time-course samples from rhythmic animals using RNAseq.

**Results**. We identified a comprehensive suite of circadian clock gene homologues in *T. saltator* including the 'core' clock genes *period* (*Talper*), *cryptochrome 2* (*Talcry2*), *timeless* (*Taltim*), *clock* (*Talclk*), and *bmal1* (*Talbmal1*). In addition we describe the sequence and putative structures of 23 clock-associated genes including two unusual, extended isoforms of pigment dispersing hormone (*Talpdh*). We examined time-course RNAseq expression data, derived from tissues harvested from behaviourally rhythmic animals, to reveal rhythmic expression of these genes with approximately circadian period in *Talper* and *Talbmal1*. Of the clock-related genes, *casein kinase IIβ* (*TalckIIβ*), *ebony* (*Talebony*), *jetlag* (*Taljetlag*), *pigment dispensing hormone* (*Talpdh*), *protein phosphatase 1* (*Talpp1*), *shaggy* (*Talshaggy*), *sirt1* (*Talsirt1*), *sirt7* (*Talsirt7*) and *supernumerary limbs* (*Talslimb*) show temporal changes in expression.

**Discussion**. We report the sequences of principle genes that comprise the circadian clock of *T. saltator* and highlight the conserved structural and functional domains of their deduced cognate proteins. Our sequencing data contribute to the growing inventory of described comparative clocks. Expression profiling of the identified clock genes illuminates tantalising targets for experimental manipulation to elucidate the molecular and cellular control of clock-driven phenotypes in this crustacean.

Corresponding author
David C. Wilcockson,
dqw@aber.ac.uk

## INTRODUCTION

Nearly all organisms are exposed to monotonous cyclic alterations in their environment. Competence to gate behaviour and physiology in tune with these changes is orchestrated by so-called circadian clocks, the cardinal features of which are: the clock mechanism is autonomous and endogenous i.e., it persists in the absence of external cues; the period of the rhythm is temperature compensated and is entrained by relevant cyclic environmental cues. For example, in the terrestrial realm, organisms invariably exhibit daily activity rhythms in temporal correspondence to the light and dark of day and night. Our understanding of the molecular and cellular underpinnings of circadian clocks has advanced tremendously in recent years and comparative studies have benefitted from the advent of next generation sequencing strategies to reveal clock genes in non-model species, including crustaceans (*Christie et al., 2013*; *De Pitta et al., 2013*; *Nesbit & Christie, 2014*; *Toullec et al., 2013*).

In the fruit fly brain about 150 cells, primarily in the protocerebrum and optic lobe, contribute to the central clock oscillatory system, although this number varies throughout arthropods. Indeed, in some Lepidoptera (*Sauman et al., 2005*) and Crustacea (*Beckwith et al., 2011*; *Zhang et al., 2013*) this number seems to be much lower. The consensus arthropod model of the central oscillatory mechanism is based on transcriptional/translational feedback loops (TTFLs), with positive loops driving the expression of negative factors that feed-back to inhibit the positive loops and hence their own transcription. Despite interspecific variation, the principle mechanisms and components are conserved across diverse taxa (For reviews, see *Allada & Chung, 2010*; *Hardin, 2009*; *Sandrelli et al., 2008*). Briefly, in *Drosophila* the transcription factors CLOCK and CYCLE (CLK and CYC) drive the expression of *period* (*per*) and *timeless* (*tim*). The translation of *per* and *tim* throughout the evening and early night results in the cytoplasmic accumulation of their cognate proteins that subsequently form stable heterodimers and translocate to the nucleus where they interfere with the action of CLK and CYC to inhibit their own transcription. Under constant conditions such as DD, this autoregulatory feedback loop takes circa 24 h to complete due to the opposing phosphorylation and dephosphorylation actions of various casein kinases and phosphatases that alter circadian protein stability. For example, the stability of PER is facilitated by casein kinase 1$\varepsilon$ (CK1$\varepsilon$) or DOUBLETIME (DBT) phosphorylation and translocation of the PER/TIM dimer is mediated by phosphorylation via glycogen synthase kinase $\beta$ or "SHAGGY" (SGG). In fruit flies maintained under LD conditions the blue light activated flavoprotein cryptochrome (dCRY1) acts as a transduction pathway to relay photic stimuli to the TTFL. When photo-activated at dawn dCRY1 facilitates the degradation of TIM, via the CRYPTOCHROME/JETLAG (JET) complex. This consequently leads to destabilisation of PER, which is ubiquitinated by E3 ubiquitin ligase SUPERNUMERARY LIMBS (SLIMB), and degraded via the proteasome. The resultant de-repression of dCLOCK and dCYCLE 'resets' the clock at dawn synchronising (or entraining) the system to the ambient conditions. Protein phosphatases (PP) including PP2A and PP1 regulate these phosphorylation events and so their action contributes to the temporal control of the feedback loops. A secondary negative loop also governs precisely timed oscillations; the CLK-CYC heterodimer initiates the transcription of the basic zipper (bZip) activator

PAR domain protein 1ε (PDP1ε) and bZip transcriptional repressor *vrille* (*vri*) (*Blau & Young, 1999*) by binding E-boxes in the *vri* promoter. VRI accumulation culminates in the repression of *clk* transcription by binding VRI/PAR domain protein 1ε (PDP1ε) promoters (*Cyran et al., 2003*). VRI dependent repression also mediates transcription of output mRNAs that cycle in phase with *clk*, such as *cryptochrome* (*Glossop et al., 2003*). In mammals and some arthropods, CRYPTOCHROME 2 (CRY2) acts as a negative repressor of the CLK/BMAL positive loop. In flies the actions of CLK/CYC E-boxes are also targeted by a basic-helix-loop-helix repressor, CLOCKWORK ORANGE (CWO), which is thought to sustain rhythmicity (*Kadener et al., 2007*; *Lim et al., 2007*; *Matsumoto et al., 2007*).

In insects, the octadecapeptide pigment dispersing factor (PDF) that occurs in subsets of clock neurons plays pivotal roles in synchronising oscillator neurons in the clock network. Perturbation of PDF function leads to arrhythmicity or incompetence to adjust phase and period of rhythmic phenotypes to changing environmental cues (*Shafer & Yao, 2014*). PDF, or rather the orthologous pigment dispersing hormone (PDH) was first discovered in crustaceans where they play a neurohormonal role as well as a neuromodulatory function, effecting dispersion of retinal pigments and integumental chromatophores, including the diurnal rhythm of this action. In crustaceans, PDHs occur in several isoforms and the neuroarchitecture of PDH cells is more complex than for PDF, suggesting additional functions to pigment dispersion. However, the defined role(s) for PDH in rhythmic phenotypes of crustaceans remains poorly described (*Beckwith et al., 2011*; *Strauss & Dircksen, 2010*; *Wilcockson et al., 2011*)

*Talitrus saltator* lives in the sandy substratum of the supralittoral zone of beaches on the European Atlantic coast and in the Mediterranean. During the day it remains buried in the sand, emerging after dark to make foraging excursions along the land-beach axis. Before dawn it relocates to its burrowing zone to seek refuge, reinterred in the sand. This circadian locomotor activity persists in the absence of external cues and is thus endogenously driven (*Bregazzi & Naylor, 1972*; *Edwards & Naylor, 1987*). Remarkably, *T. saltator* also maintains its preferred position on the shore by orientating using visual cues such as the sun and moon as a compass guide (*Pardi & Papi, 1952*; *Ugolini, 2003*; *Ugolini et al., 1999*; *Ugolini, Melis & Innocenti, 1999*; *Ugolini et al., 2007*). This capability necessitates compensation for azimuthal changes over time (*Ugolini, Tiribilli & Boddi, 2002*). Thus, these clock-controlled phenotypes contribute to the fitness of *T. saltator*.

The well-defined behavioural phenotypes of *T. saltator* enable a range of comparisons with other organisms to be drawn. For example, another peracarid crustacean, the isopod *Eurydice pulchra* possesses independent circatidal and circadian clocks (*Zhang et al., 2013*) and the staggering time-compensated navigational mechanisms of the monarch butterfly (*Reppert, 2006*) are comparable to those of *T. saltator*. This brings the relevance of comparative clock biology sharply into focus; detailed analysis of a diverse non-model species may reveal commonalities or differences that give insight into how each has evolved and functions. Thus, *T. saltator* represents an excellent, tractable model for time-compensated orientation.

Despite decades of elaborate behavioural analyses, the molecular basis of clock-driven behavioural phenotypes in *T. saltator* is not understood, even at the most fundamental

level. Therefore, we sought to elucidate the sequences of canonical clock genes and their temporal expression dynamics in *T. saltator* to provide a platform from which to explore the neuromolecular mechanisms of circadian clock mediated phenotypes in this animal. This work contributes also to the steadily growing inventory of crustacean transcriptomes and is applicable to further exploration of comparative chronobiology and animal orientation in a tractable and ecologically important model.

## MATERIALS AND METHODS

### Animal husbandry, tissue sampling and RNA extraction

Animals were caught by hand from Ynyslas beach, Wales, UK at night during their active period and returned to glass tanks containing damp sand at ambient temperature (17 °C) and a 12:12 LD regime. Lighting at dawn and dusk was ramped/dimmed over 30 min to approximate natural conditions. Animals were fed fish food flakes *ad libitum*. After seven days acclimation, the rhythmic emergence and locomotor activity of a sub-sample of 60 animals was established. *T. saltator* has been shown to express more robust and less variable locomotor rhythms in small groups (*Bregazzi & Naylor, 1972*). Therefore, animals were housed in groups of five in a glass tank containing 10 cm-deep damp sand and compartmentalized with Plexiglas dividers. Across each compartment, infrared beams were passed via bespoke recording apparatus fabricated by Trikinetics (Waltham, MA, USA). All activity, registered as interruptions to infrared beams was recorded via proprietary software on a stand-alone PC. Activity data were recorded in one minute bins and was analysed and plotted using ClockLab software (Actimetrics, Wilmette, IL, USA) run via Matlab® v6.2. *T. saltator* brains were rapidly dissected from animals drawn from the population shown to be behaviourally rhythmic (see above). Dissections were done in ice-chilled DEPC treated physiological saline. Ten brains were pooled to form each replicate at 3-hour intervals and snap frozen in liquid nitrogen before storing at −80 °C until use.

Total RNA was extracted using Trizol® (Invitrogen ™, Thermo Fisher, UK) according to the manufacturer's instructions except that an additional wash step in 75% ethanol of the RNA pellet was introduced prior to drying and rehydration in 30µl DEPC-treated water. All RNA samples were treated with Turbo DNA-*free* ™(Ambion®, Thermo Fisher, Hampshire, UK) to remove contaminating DNA. RNA was then pooled for degenerate PCR, RACE PCR and Illumina HiSeq2500 sequencing, whilst RNA time-course samples were prepared separately for temporal expression analysis.

### Degenerate PCR

Initially we used a strategy of degenerate PCR and 5′ and 3′ Rapid amplification of cDNA ends (RACE) to identify full-length core canonical clock genes. Full details of the PCR conditions used can be found in File S1 and primers used are tabulated in Table S1.

### Illumina RNAseq protocol

The TruSeq cDNA library preparation protocol (Illumina, Cambridge, UK) was carried out on eight time-point samples according to manufacturer's instructions. Amplified cDNA was run on a 1% agarose gel to validate correct library size range. The cDNA libraries were

sequenced on an Illumina RNA TruSeq sequencer (Illumina Inc.) and quality checked at Aberystwyth University's Core Genomics Facility. Read library quality was investigated using FastQC (http://www.bioinformatics.babraham.ac.uk/projects/fastqc/), thereafter all samples were pooled and *de novo* transcriptome assembly was performed with Trinity software, version 2012-10-05 (*Grabherr et al., 2011*). Transcripts smaller than 300 bases were removed from the pooled assembly. This consisted of 186,495 contigs with a mean length of 1,215 bp pairs and an assembly N50 statistic of 2,371 bp.

## Transcriptome mining and sequence analysis

Assembled contigs were initially interrogated for canonical clock gene transcripts using the programme BioEdit (*Hall, 1999*), Basic local alignment (BLAST) searches (http://blast.ncbi.nlm.nih.gov/Blast.cgi) were done using *Drosophila melanogaster* protein sequences as search terms and tblastn search functions set with default parameters (*e*-value threshold at $1e^{-100}$ except for PDH, where stringency was relaxed to 1000). Where no sequence information was available for *Drosophila* (e.g., cryptochrome 2) alternative arthropod sequences were used, details of which are given in Table 1. Contigs were translated using EXPASY Translate tool (http://web.expasy.org/translate/) and the coding sequences checked manually using blastP for homology to known circadian proteins.

EMBL SMART (*Letunic, Doerks & Bork, 2009*) servers were used to detect and analyse the conserved functional domains and motifs of clock proteins. Deduced *T. saltator* protein sequences were used to query the FlyBase (http://flybase.org) and NCBI non-redundant protein databases using the blastp algorithm. The identity and similarity between protein sequences were calculated using EMBOSS Pairwise Alignment Algorithms (http://www.ebi.ac.uk). The server SignalP v.4 (*Petersen et al., 2011*) was used to predict the presence and location of signal peptide cleavage sites.

## Quantitation of transcript abundance

Using the Trinity downstream analysis tools, reads from each time-point were mapped to the pooled assembly and transcript abundance estimation was carried out using RSEM (*Li & Dewey, 2011*). In this way the FPKM (Fragments Per Kilobase of exon per Million fragments mapped) values were calculated for each time point. Following this TPM values (transcripts per million) were calculated following *Wagner, Kin & Lynch (2012)*. Quantitative PCR methods for TPM validation are given in File S1 and primers in Table S1. All data files for time-course RNAseq can be found on the NCBI SRA data base, bioproject Accession No. 297565.

## Blast2GO analysis- transcriptome annotation

Before running Blast2GO, the number of transcripts was further reduced by clustering them using CD-HIT-EST (version 4.5.4) with default options except for the sequence identity threshold, which was set to 95% (*Li & Godzik, 2006*; *Li, Jaroszewski & Godzik, 2001*). The clustering step removes very similar splice variants, which may include some misassemblies, and it significantly reduces the number of transcripts to be included in the time-consuming BLAST analysis. In total 156,766 transcripts with a mean length of 1,534 bp and an assembly N50 statistic of 968 bp were run through BLAST. Using the

**Table 1** Identified putative *Talitrus saltator* circadian protein-encoding transcripts.

| Clock gene query protein | Transcriptome search sequence Accession No. (all *D. melanogaster*) | *Talitrus* transcript/protein identifications | | | | | |
|---|---|---|---|---|---|---|---|
| | | Transcript | | | Protein | | |
| | | Trinity contig ID | Length[a] | RACE | Name | CDS | Length[b] |
| *Core clock proteins* | | | | | | | |
| CRY2 | n/a[c] | comp100937_c0_seq1[c] comp102609_c0_seq3[c] | 1,843 | 5′ + 3′ | Tal-CRY2 | Full | 565 |
| CLOCK | AAC62234 | comp100688_c1_seq1 | 5,723 | 3′[d] | Tal-CLK | Partial | 1,907 |
| PERIOD | AAF45804 | comp102279_c0_seq7 | 8,001 | | Tal-PER | Full | 1,557 |
| TIMELESS | AAC46920 | comp849619_c0_seq1 | 1,209 | | Tal-TIM | Partial | 402 |
| CYCLE | AAF49107 | comp12103_c0_seq1 | 1,807[e] | | Tal-BMAL1 | Full | 602 |
| | | comp939723_c0_seq1 | | | | | |
| *Clock-associated proteins* | | | | | | | |
| PDH | n/a[f] | comp92607_c0_seq2 | 2,471[g] | 5′ + 3′[d] | Tal-PDH I | Full | 129 |
| | | comp97165_c0_seq3 | 3,392 | | Tal-PDH II | Full | 89 |
| CASEIN KINASE 2 $\alpha$ | AAN11415 | comp102480_c0_seq1 | 5,147 | | Tal-CK2 $\alpha$ | Full | 353 |
| CASEIN KINASE 2 $\beta$ | AAF48093 | comp99101_c0_seq3 | 1,567 | | Tal-CK2 $\beta$ | Full | 220 |
| CLOCKWORK ORANGE | AAF54527 | comp1009591_c0_seq1 | 503 | | Tal-CWO | Partial | 167 |
| DOUBLETIME (or CK1 $\varepsilon$) | AAF57110 | comp87763_c0_seq1 | 1,092 | | Tal-DBT | Full | 310 |
| PDP1 $\varepsilon$ | AAF04509 | comp98345_c0_seq1 | 3,423 | | Tal-PDP1 $\varepsilon$ | Full | 508 |
| PP1 | CAA39820 | comp97405_c0_seq1 | 1,725 | | Tal-PP1 | Full | 357 |
| PP2A –subunit MICROTUBULE STAR | AAF52567 | comp98380_c0_seq1 | 2,981 | | Tal-MTS | Full | 309 |
| PP2A –subunit WIDERBORST | AAF56720 | comp102157_c1_seq1 | 2,474 | | Tal-WBT | Full | 458 |
| PP2A –subunit TWINS | AAF54498 | comp99704_c0_seq3 | 1,633 | | Tal-TWS | Full | 445 |
| SHAGGY | AAN09084 | comp99811_c0_seq7 | 4,413 | | Tal-SGG | Full | 418 |

**Table 1** (*continued*)

| Clock gene query protein | Transcriptome search sequence Accession No. (all *D. melanogaster*) | *Talitrus* transcript/protein identifications | | | | | |
|---|---|---|---|---|---|---|---|
| | | Transcript | | | Protein | | |
| | | Trinity contig ID | Length[a] | RACE | Name | CDS | Length[b] |
| SUPERNUMERARY LIMBS | AAF55853 | comp98870_c0_seq1 | 2,121 | | Tal-SLIMB | Full | 588 |
| VRILLE | AAF52237 | comp100474_c0_seq6 | 3,949 | | Tal-VRI | Full | 509 |
| EBONY | AAF55870 | comp99283_c0_seq2 | 4,380 | | Tal-EBONY | Full | 974 |
| RORA | NP_788301 | comp99654_c0_seq3 | 2,217 | | Tal-RORA | Partial | 599 |
| REVERB | NP_730321 | comp101252_c0_seq2 | 5,385 | | Tal-REVERB | Full | 1,110 |
| SIRT1 | NP_477351 | comp101818_c1_seq1 | 4,033 | | Tal-SIRT1 | Partial | 955 |
| SIRT2 | NP_650880 | comp97450_c0_seq4 | 2,275 | | Tal-SIRT2 | Partial | 376 |
| SIRT4 | NP_572241 | comp92313_c0_seq2 | 2,977 | | Tal-SIRT4 | Partial | 354 |
| SIRT6 | NP_649990 | comp69157_c0_seq1 | 1,209 | | Tal-SIRT6 | Partial | 402 |
| SIRT7 | NP_651664 | comp95761_c0_seq1 | 5,180 | | Tal-SIRT7 | Partial | 948 |
| JETLAG | NP_608880 | comp100423_c0_seq4 | 2,454 | | Tal-JET | Partial | 458 |

**Notes.**

[a] length in nucleotides.

[b] length in amino acids.

[c] Homology cloning sequence used to search transcriptome. Trinity contigs likely from misassembly, other gene details stated taken from homology cloning and RACE PCR.

[d] Trinity contig extended 3′ sequence past RACE-derived sequence, other gene details stated taken from contig sequence.

[e] Overlapping contigs combined.

[f] *Uca pugilator* octadecapeptide $\beta$-PDH consensus sequence used.

[g] Includes combined contig sequence and overlapping 5′ RACE sequence.

BLAST output the transcripts were annotated using BLAST2GO software (*Conesa et al., 2005*; *Gotz et al., 2008*) including reports for Gene Ontology (GO) terms and EC numbers for the KEGG pathway. The BLAST2GO cut-off parameters used to filter out poor quality BLAST hits for the annotation were as follows: Annotation rule cut-off = 55; $E$-value = 1e–6; Hit-HSP overlap = 0; and the GO weight = 5.

## Determining cycling transcripts

Rhythmicity in clock gene expression was determined using the JTK_CYCLE software (http://openwetware.org/wiki/HughesLab:JTK_Cycle) developed by Professors Michael E. Hughes, Karl Kornacker and John Hogenesch (*Hughes, Hogenesch & Kornacker, 2010*; *Miyazaki et al., 2011*) following the JTK_CYCLE Users guide. Changes in gene expression values (TPM) were also analysed by one-way ANOVA.
## Data availability

Raw data files have been deposited in public sequencing databases as indicated in the text or at https://figshare.com/s/a2513243c63bf557b720.

# RESULTS

Animals entrained under 12:12LD regimes showed robust activity rhythms when released into constant darkness (DD) with peak activity occurring in the middle of the subjective night (Fig. 1A). Periodogram analysis of representative animals revealed a period (tau) of 24.15 h (Fig. 1B). The demonstration of rhythmicity in these animals was essential to our sampling strategy; tissue for gene discovery was taken across one complete daily cycle and pooled to ensure capture of transiently expressed transcripts. Additionally, expression profiling of transcriptome contigs required samples harvested from rhythmic animals in-phase with respect to each other.

### Identification of putative *Talitrus saltator* circadian proteins

Our principle objective was to describe the cDNAs encoding the core elements of the circadian clock system in *T. saltator*, initially adopting homology cloning and standard sequencing approach but, subsequently superseded by RNAseq strategies. The Illumina RNAseq generated 141,769,456 reads, 128,386,193 of these were assembled into 156,766 clustered contigs (minimum length 300 bp). This *T. saltator* Transcriptome Shotgun Assembly project has been deposited at DDBJ/EMBL/GenBank under the Accession No. GDUJ00000000. The version described in this paper is the first version, GDUJ01000000. From this transcriptome we identified contigs encoding putative circadian clock genes (Table 1). The coding regions and conserved functional domains of identified 'core' clock genes *period* (*Talper*), *timeless* (*Taltim*), *cryptochrome2* (*Talcry2*) and *bmal1* (*Talbmal1*) and *clock* (*Talclk*) are diagrammatically represented in Fig. 2. In the interests of space, alignments of all genes (core and clock-related) are shown in Figs. S1–S26 and all BLAST results are tabulated in Tables S2–S4.

### Cryptochrome 2

Degenerate PCR coupled with RACE PCR yielded a 1,843 bp cDNA sequence coding for a 565 amino acid protein; a putative cryptochrome 2 assigned TalCRY2. An identical sequence was revealed in the transcriptome data but assembled as two separate contigs (Table 1). High fidelity PCR amplification and sequencing confirmed that *Talcry2* is expressed as one contiguous transcript. The deduced protein sequence contains two SMART identified domains, a DNA photolyase domain and a FAD binding 7 domain (Figs. 2 and S1) and shared an identity of 46.4% and a similarity of 56.0% with *Danaus plexippus* CRY2 (Accession No. ABA62409; *Zhu et al., 2005*). The highest sequence identity protein in the NCBI non-redundant database is that for the closely related isopod crustacean, *Eurydice pulchra* CRY2 sequence (Accession No. AGV28717; *Zhang et al., 2013*).

### Period and timeless

For *Talper* we mined an 8,001 bp contig encoding a putative full-length protein of 1557 amino acids aligning to dPER with an identity of 29% and a similarity of 50%. The deduced

**A**

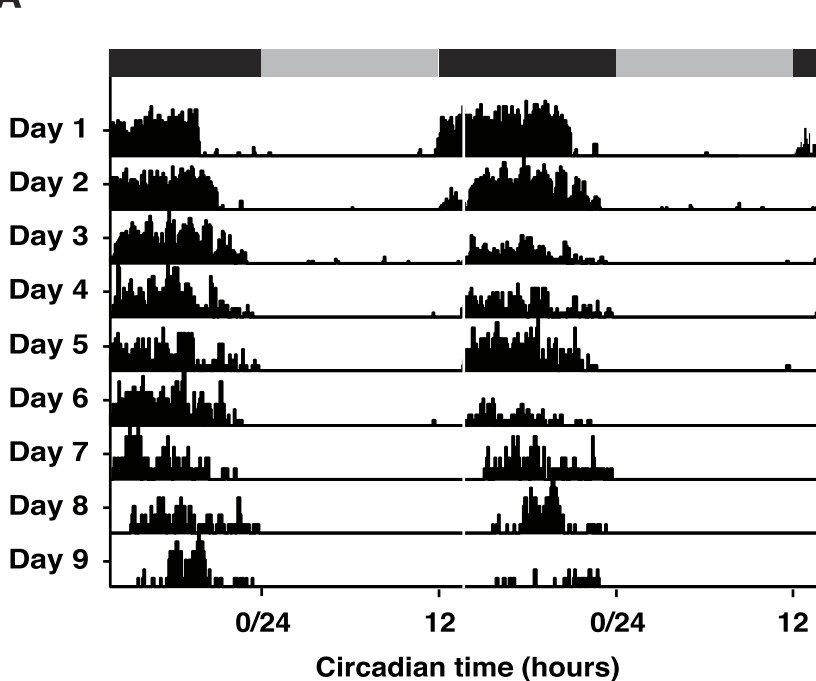

**B**

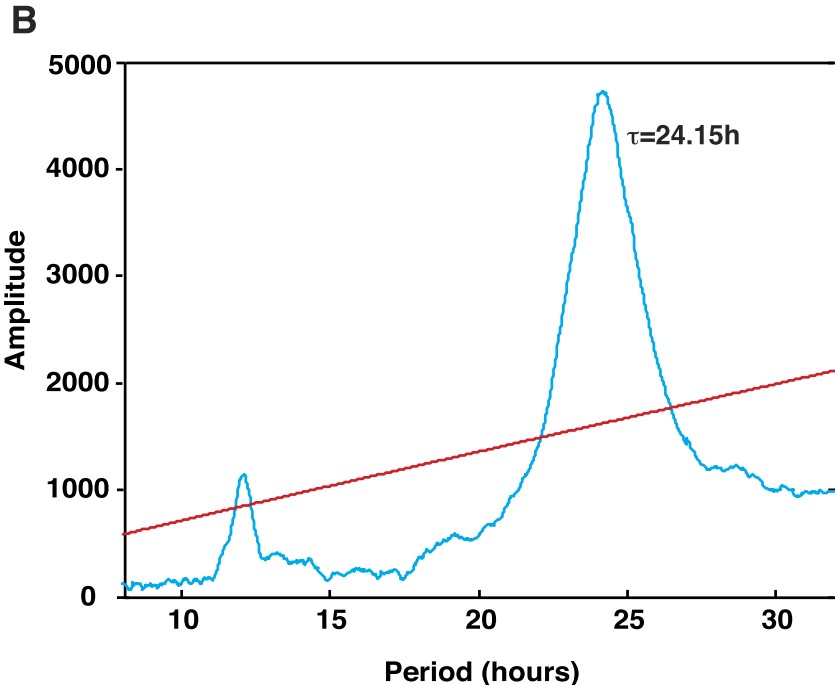

**Figure 1** *Talitrus saltator* **exhibits free-running circadian locomotor activity rhythms.** (A) Plots of activity in five representative animals removed from the shore and held in DD over 9 days. Grey and black bars show time of subjective night and day, respectively. (B) Chi Square Periodogram analysis of activity data of five representative animals recorded in DD over nine days. The period of activity ($\tau$) is shown inside the plot. Red line represents significance at $P < 0.001$.

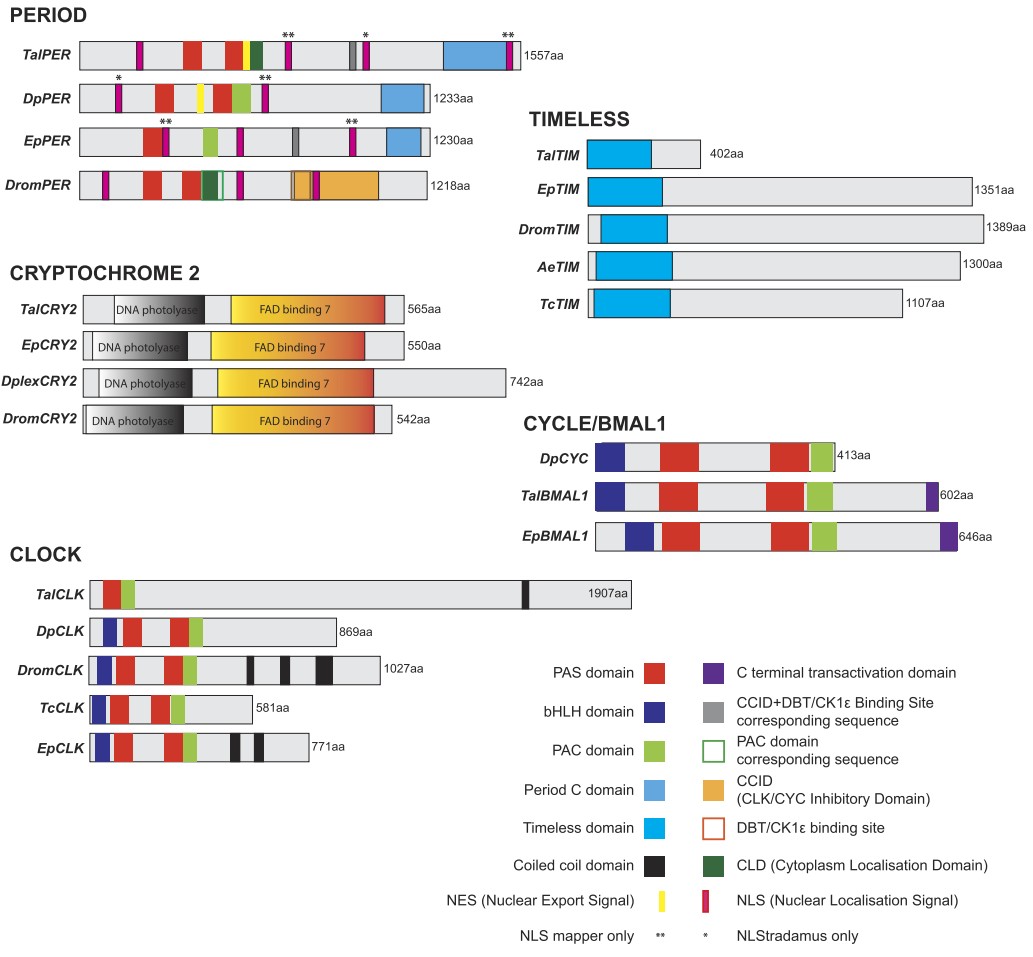

**Figure 2** **Cartoon representing conserved domains of core clock gene proteins.** Putative *Talitrus saltator* core clock protein amino acid sequences aligned with other arthropod circadian clock protein sequences. SMART identified domains are coloured (see key for colours corresponding to domains). *Tal*, *Talitrus saltator*; *Dp*, *Daphnia pulex*; *Ep*, *Eurydice pulchra*; *Drom*, *Drosophila melanogaster*; *Dplex*, *Danaus plexippus*; *Tc*, *Tribolium castaneum*; *Ae*, *Aedes aegypti*.

TalPER contains characteristic features of PER proteins including two PAS domains and a PAC domain as well as nuclear export signal, cytoplasmic localization and nuclear localization signal motifs. In common with the isopod *E. pulchra*, TalPER also has a CK1ε binding site sequence and Period C domain (Figs. 2 and S2). The putative TalPER sequence aligned most closely to dPER-PB in FlyBase (FlyBase No. FBpp0304590) and in the NCBI database to *Eurydice pulchra* PER (Accession No. AGV28714; *Zhang et al., 2013*). We also used the *Drosophila melanogaster* TIM (dTIM) protein (Accession No. AAC46920; *Myers et al., 1995*) as a search term to identify a 1,209 bp contig with a coding region for a 402 amino acid protein sharing sequence similarity to dTIM (identity of 24.3% and similarity of 41.0%, Fig. S3). SMART analysis of this sequence identified a 228 amino acid Timeless domain (Figs. 2 and S3). The partial protein sequence, assigned TalTIM aligned most closely in the NCBI database to the partial Timeless-like protein of the termite *Zootermopsis nevadensis*

(Accession No. KDR17447; *Terrapon et al., 2014*) and in FlyBase to Timeout-PA (FlyBase No. FBpp0082180).

## Clock

Using the *D. melanogaster* CLK (dCLK) sequence (Accession No. AAC62234; *Bae et al., 1998*) as search term, a single 5,723 bp contig was identified as a candidate TalCLK encoding sequence. However, this sequence lacked a stop codon and we were unable to deduce the C-terminal end of the protein, which is missing a C-terminal transactivation domain (Figs. 2 and S4). Nevertheless, our partial sequence contains SMART identified domains including one PAS domain, one PAC domain and coiled coil domains, whilst missing a bHLH domain and a second PAS domain present in dCLK. The TalCLK PAS domain is likely homologous to the PAS-2 domain of dCLK as it aligned better (identity of 64.2% and a similarity of 80.6%) than it did to the PAS-1 domain (identity of 17.3% and a similarity of 36.0%). Moreover, the PAC domains align closely with an identity of 75.0% and a similarity of 90.9% to that of dCLK. The most significant hit for TalCLK in the NCBI database was CLK of the prawn *Macrobrachium rosenbergii* (Accession No. AAX44045; *Yang et al., 2006*). In FlyBase the most similar protein is the *D. melanogaster* CLK orthologue (FlyBase No. FBpp0306710).

## Cycle/Bmal1

By searching the transcriptome using the *Drosophila melanogaster* CYC (dCYC) sequence (Accession No. AAF49107; *Adams et al., 2000*), we identified two contigs as potential TalCYC/BMAL encoding sequences each of which aligned closely to separate regions of dCYC (Figs. 2 and S5). In fact these two contigs, when correctly oriented, overlap to form one contiguous sequence encoding a characteristic bHLH domain, two PAS domains a PAC and a C terminal transactivation domain. This sequence was confirmed by standard, Hi-fidelity PCR, cloning and sequencing. TalBMAL1 aligned with identity of 43% and similarity of 57% to BMAL1 of signal crayfish *Pacifasticus leniusculus* (Accession No. AFV39705). TalBMAL1 aligned with 44% identity 58% similarity to *E. pulchra* BMAL1.

In addition to the 'core' clock elements, we also isolated a comprehensive suite of 'clock-associated' transcripts from contigs that conceptually translate to proteins known to contribute to circadian clock functioning (Table 1). In the interests of space, we refer the reader to the Supplemental Information; herein the majority of detail is omitted from the text. Sequence alignments are displayed in Figs. S6–S26 whilst BLAST search outputs of identified contig sequences to nucleotide, protein and FlyBase repositories are given in Tables S2–S4.

## Casein Kinase II (CKII) α and β subunits

We identified single contigs coding for putative CK11α and CKIIβ. For CK11α we found a 5,147 bp contig enveloping a coding sequence for a 353 amino acid protein including a highly conserved serine/threonine protein kinase catalytic domain (Table 1; Fig S6). This putative TalCKIIα sequence aligned in the NCBI database most closely to human CK2 Chain A (Accession No. 1NA7_A; *Niefind et al., 1998*) and CKIIα from the copepod crustacean, *Paracyclopina nana* (Accession No. AII16523). For CKIIβ we identified a

1,567 bp coding sequence for a 220 amino acid protein, including a casein kinase regulatory subunit domain (Table 1; Fig. S6). This sequence aligned most closely in the NCBI database to parasitic wood wasp *Orussus abietinus* CKIIβ (Accession No. XP_012287730).

## Clockwork orange (CWO)

A candidate *cwo* contig was mined that encodes a partial protein sequence of 167 amino acids (Table 1; Fig. S7) and included a SMART-identified bHLH domain and the Orange of the Hairy/E (SPL) family domain. TalCWO matched most closely in the NCBI database to an uncharacterized protein from the mite *Metaseiulus occidentalis* (Accession No. XP_003744690); however, the same sequence aligned with CLOCKWORK ORANGE from the firebug *Pyrrhocoris apterus* (Accession No. AGI17571; *Bajgar, Jindra & Dolezel, 2013*) with an *e* value of $8e^{-42}$.

## Doubletime (DBT)

A putative 1,092 bp *dbt* contig coding for a deduced a full-length 310 amino acid protein was identified containing a serine/threonine protein kinase catalytic domain (Table 1; Fig. S8). The deduced sequence aligned most closely in the NCBI database to DBT of the isopod crustacean *Eurydice pulchra* (Accession No. AGV28719; *Zhang et al., 2013*) and in FlyBase to the *Drosophila melanogaster* orthologue DCO (FlyBase No. FBpp0306615).

## Par domain protein 1ε (PDP1ε)

A single putative 3,423 bp *pdp1ε* contig with a coding sequence for a 508 amino acid protein was identified containing a basic leucine zipper domain in the C-terminal region (Table 1; Fig. S9). The deduced TalPDP1ε protein sequence aligned most closely in the NCBI database to the hepatic leukaemia factor of the clonal raider ant *Cerapachys biroi* (Accession No. EZA50108; *Oxley et al., 2014*); however PDP1ε from the mosquito *Culex quinquefasciatus* (Accession No. XP_001865130) was also a top ranking BLAST hit aligned with an e value of $1e^{-35}$. In FlyBase the sequence aligned most closely to *Drosophila melanogaster* PDP1 (FlyBase No. FBpp0289727).

## Protein phosphatase 1 (PP1)

A 1,725 bp contig was identified coding for a 357 putative PP1 containing a protein phosphatase 2Ac catalytic domain of the serine/threonine phosphatase family (Table 1; Fig. S10). The translated sequence aligned most closely to PP1 of the jumping ant *Harpegnathos saltator* (Accession No. EFN86649; *Bonasio et al., 2010*) in the NCBI database. In FlyBase the sequence aligned most closely to *Drosophila melanogaster* PP1 (FlyBase No. FBpp0306442).

## Protein phosphatase 2A (PP2A)
### PP2A catalytic subunit "MICROTUBULE STAR" (MTS)

The *Drosophila melanogaster* MTS sequence (dMTS, Accession No. AAF52567; *Adams et al., 2000*), was used as a search term to identify a contig 2,981 bp in length and coding for a putative 309 orthologue to dMTS with an identity of 91.9% and a similarity of 96.8% (Table 1; Fig. S11). The putative TalPP2A includes a protein phosphatase 2Ac catalytic domain of the serine/threonine phosphatase family with an identity of 93.4% and

a similarity of 98.2% to the fruit fly orthologue. The closest match in the NCBI database was the putative serine/threonine protein phosphatase PP-V from the body louse *Pediculus humanus corporis* (Accession No. XP_002426726). The closest match in FlyBase was the *Drosophila melanogaster* MTS homologue (FlyBase No. FBpp0310063).

### PP2A catalytic subunit "WIDERBORST" (WBT)

We identified a 2,474 bp sequence that codes for a putative 458 amino acid protein. The deduced protein sequence, although 66 amino acids shorter than that of *Drosophila* WBT (the TalWBT protein lacks a sequence at the C terminal end) aligned with the fruit fly orthologue (identity of 73.8%) % (Table 1; Fig. S12). In common with other species, the *T. saltator* candidate contains B56 domains (serine/threonine-protein phosphatase 2A 56 kDa regulatory subunit epsilon). The deduced TalWBT protein aligned most closely in the NCBI database to serine/threonine-protein phosphatase 2A 56 kDa regulatory subunit epsilon of the red flour beetle *Tribolium castaneum* (Accession No. XP_971164) and in FlyBase to *Drosophila melanogaster* WBT (FlyBase No. FBpp0084575).

### PP2A catalytic subunit TWINS (TWS)

A unique 1,633 bp contig coding for a putative 445 amino acid protein with high identity to *Drosophila* TWS, (Table 1; Fig. S13) was identified. The deduced sequence contained seven WD40 domains identified using SMART. The protein aligned most closely in the NCBI database to PP2A subunit B of the mud crab *Scylla paramamosain* (Accession No. AFK24473) and to *Drosophila melanogaster* TWINS (FlyBase No. FBpp0081671).

### Shaggy (SGG)

A 4,413 bp transcript incorporating a coding region for a 418 amino acid protein that contained a SMART identified serine/threonine protein kinase catalytic domain (Table 1; Fig. S14). In the NCBI database, the sequence aligned most closely to glycogen synthase kinase-3 of the turnip sawfly *Athalia rosae* (Accession No. XP_012256017) and in FlyBase to *Drosophila melanogaster* SGG (FlyBase No. FBpp0070450).

### Supernumerary limbs (SLIMB)

A candidate 588 amino acid putative SLIMB protein encoded within a 2,121 bp contig was mined. This sequence included one D domain of beta-TrCP, one F box and seven WD40 domains (Table 1; Fig. S15). The deduced sequence aligned in the NCBI database most closely to F-box/WD repeat-containing protein 1A of the termite *Zootermopsis nevadensis* (Accession No. KDR19729; *Terrapon et al., 2014*) and in FlyBase to *Drosophila melanogaster* SLIMB (FlyBase No. FBpp0303082).

### Vrille (VRI)

A single 3,949 bp contig incorporating coding sequence for a 509 amino acid protein was identified as a putative *vrille* transcript. The putative TalVRI protein contains a SMART identified basic region leucine zipper domain (Table 1; Fig. S16) and aligned most closely in the NCBI database to nuclear factor interleukin-3-regulated protein of the termite *Zootermopsis nevadensis* (Accession No. KDR19729; *Terrapon et al., 2014*) and in FlyBase to *Drosophila melanogaster* VRI (FBpp0312171).

## Ebony

A single 4,380 bp contig coding for a putative 974 amino acid protein with one AMP binding domain, one AMP binding C domain and one PP binding domain was mined (Table 1; Fig. S17). The deduced protein most closely aligned in the NCBI database to the $\beta$-alanyl conjugating enzyme of the cockroach *Periplaneta americana*. *P. americana* EBONY (Accession No. CAI26307; *Blenau & Baumann, 2005*) has been shown to have $\beta$-alanyl-dopamine (DA) synthase (BAS) enzymatic activity and the *Drosophila melanogaster* EBONY sequence (Accession No. ABO27280) aligned with the TalEBONY candidate sequence with an e value of $8e^{-90}$, supported by EBONY (FlyBase No. FBpp0083505) being the most closely aligned protein to the TalEBONY query in FlyBase.

## Pigment dispersing hormone 1 and II (PDHI and PDHII)

In the Trinity assembly, a 2,430 bp contig was identified by searching for the conserved NSELINS domain. The identified contig included a coding region for a putative 129 amino acid TalPDH prepropeptide (TalPDH-I, Table 1, Fig. S18). This was subsequently extended to 2,471 bp by the addition of a 5′ UTR by RACE PCR. The TalPDH-I contains a signal peptide between residues 20–21, a 77 amino acid PDH-precursor-related peptide (PPRP) sequence ending in a K-R dibasic cleavage site. Unusually however, the deduced mature peptide sequence is 32 amino acids long and lacks an amidation signal. SMART identified PDH domains are present in the TalPDH. The deduced 32 amino acid Tal-PDH-I mature peptide has an identity of 40.6% and a similarity of 53.1% with the *Uca pugilator* $\beta$-PDH consensus sequence.

A second putative TalPDH encoding transcript was found in the transcriptome of 3,392 bp in length that included a coding region for an 89 amino acid prepropeptide that we assigned TalPDH-II (Table 1; Fig. S18). This prepropeptide sequence was predicted to include a 23 residue signal peptide. The 43 amino acid PPRP thus extends from residue 24 to a K-R dibasic cleavage site at residue 66. The 23 amino acid mature peptide terminates in an amidation signal and has an identity of 47.8% and a similarity of 69.6% with *Uca pugilator* $\beta$-PDH consensus sequence. No domains were identified by SMART but a corresponding PDH domain region was identified.

## RORA

One 2,217 bp partial contig, coding a 599 amino acid containing a C4 zinc finger domain was identified (Table 1; Fig. S19) as a predicted candidate for a *T. saltator* homologue of the hormone receptor RORA. The 3′ HOLI ligand binding domain present in other RORA proteins is not identified in TalRORA. It is possible that the HOLI ligand binding domain is present in the unsequenced 3′ section. The most closely aligned protein in the NCBI database is the house fly *Musca domestica* nuclear hormone receptor HR3 (Accession No. XP_011290218). The *Drosophila* hormone receptor-like in 46 protein (FlyBase No. FBpp0297438) was the closest protein to TalRORA in FlyBase.

### Reverb

A 5,385 bp full length contig was identified, coding for a putative 1110 amino acid protein containing both a C4 zinc finger domain and a HOLI ligand binding domain (Table 1; Fig. S20). The sequence aligned most closely in the NCBI database to nuclear hormone receptor E75, the non-mammalian REVERB homologue in the carpenter ant *Camponotus floridanus* (Accession No. XP_011259848). The closest aligning protein in FlyBase was the *Drosophila melanogaster* E75 protein (FBpp0297726).

### Sirt 1, 2, 4, 6 and 7

Five contig sequences of lengths 4,033 bp, 2,275 bp, 2,977 bp, 1,209 bp and 5,180 bp all coding for partial proteins of length 955, 376, 354, 402 and 948 amino acids, respectively were identified in the transcriptome as homologues for SIRTUIN proteins 1, 2, 4, 6, and 7. Each sequence contained one SMART identified SIR2 domain (Table 1 and Figs. S21–S25). TalSIRT1 aligned most closely to the trematode *Schistosoma mansoni* SIRT1 (Accession No. ABG78545) in the NCBI database and *Drosophila melanogaster* SIRT1 (FBpp0080015) in FlyBase. TalSIRT2 was most closely aligned to the red flour beetle *Tribolium castaneum* hypothetical protein (Accession No. EFA06770) in the NCBI database and *D. melanogaster* SIRT2 (FBpp0310647) in FlyBase. The putative TalSIRT4 protein is most closely aligned in the NCBI database to SIRT4 of the Asian citrus psyllid *Diaphorina citri* (Accession No. XP_008480918) and in FlyBase to the *D. melanogaster* SIRT4 (FBpp0070817). The TalSIRT6 sequence most closely aligns to the water flea *Daphnia pulex* hypothetical protein (Accession No. EFX74386) in the NCBI database and in the FlyBase database to *D. melanogaster* SIRT6 (FBpp0293897). TalSIRT7 most closely aligns in the NCBI database to the SIRT7 protein of the leafcutter bee *Megachile rotundata* (Accession No. XP_012143211) and aligns most closely in the FlyBase database to the *D. melanogaster* SIRT7 protein (FBpp0084733).

### Jetlag

One 2,454 bp contig coding a partial putative 458 amino acid protein was identified. This candidate putative protein sequence contains one F-box domain and multiple leucine-rich repeat domains (Table 1; Fig. S26). The sequence was most closely aligned in the NCBI database to the red flour beetle *Tribolium castaneum* F-box/LRR-repeat protein (Accession No. XP_008193983). In the FlyBase database the TalJET sequence aligned most closely to a *D. melanogaster* protein from the F-box and leucine rich repeat region group (FBpp0111980).

### Blast2Go analysis

Gene ontology (GO) annotations were categorised into the three groupings: 'Biological Process,' 'Molecular Function' and 'Cellular Component'. These categories of annotation were most easily displayed graphically at ontology level 2 (Fig. 3) where the most frequent Molecular Functions were "Binding" and "Catalytic activity"; the most common Biological Processes were "Cellular and Metabolic"; and the most common Cellular Components were "Cell Components" and "Organelle components." These GO term outputs were subsequently compared with results from transcriptomes previously generated and analysed

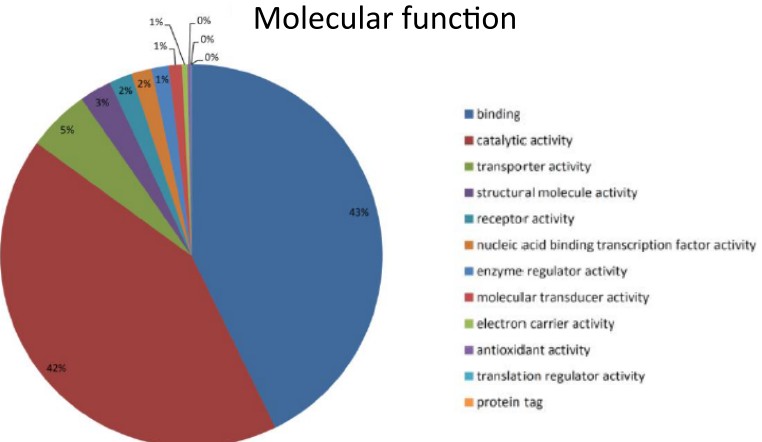

Molecular function

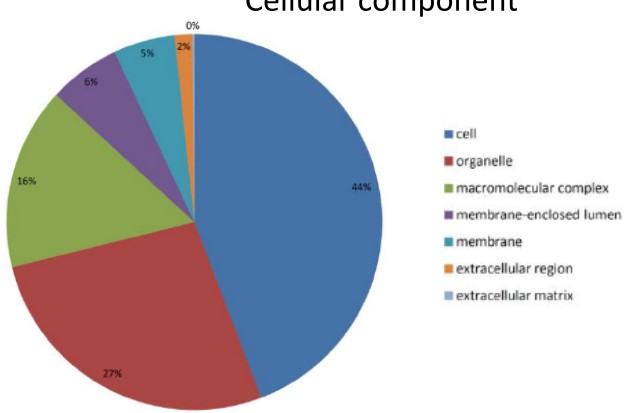

Biological process

Cellular component

**Figure 3** *Talitrus saltator* **brain tissue transcriptome BLAST2GO analysis.** Ontology level 2 data showing functional classification of the brain transcripts from *T. saltator* for three main gene ontology categories.

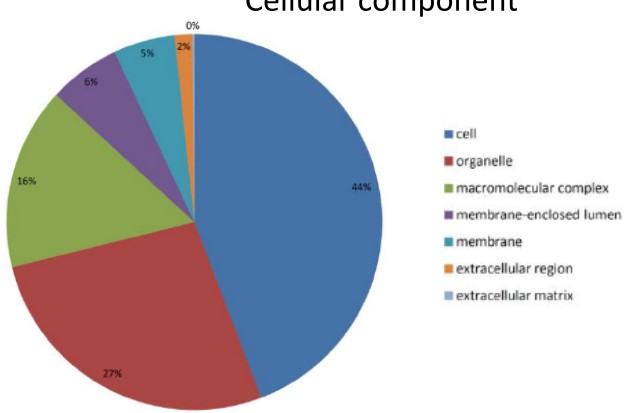

including the larva of the moth *Plutella xylostella* (*Xie et al., 2012*), the silkworm *Bombyx mori*, the parasitic nematode *Teladorsagia circumcincta* (*Menon et al., 2012*) and the North Atlantic copepod *Calanus finmarchicus* (*Lenz et al., 2014*). Proportions of GO terms were all found to be at broadly similar levels in all organisms transcriptomes and for each category analysed; for example, in the category Molecular Function ontology level 2 'binding' terms for *T. saltator*, *P. xylostella*, *B. mori* and *T. circumcincta* were 51%, 48%, 41% and 40% respectively whilst level 2 'catalytic activity' terms were 31%, 36%, 38% and 40% respectively.

**Temporal expression of clock genes**

JTK_CYCLE analysis applied to TPM values indicated oscillations (limited to a period of between 21–27h) in the core clock genes *Talper* ($q = 1.69^{-07}$) and *Talbmal1* ($q = 0.001$) (Fig. 4). Alternative analyses by ANOVA showed significant differences in mean values at each interval for *Talcry2* ($F_{7,30} = 2.745$, $P = 0.032$), *Talper* ($F_{7,30} = 4.86$, $P = 0.002$) and *Talbmal1* ($F_{7,30} = 2.445$, $P = 0.049$). Of the clock-related genes *Talpdh-II* ($q = 5.43^{-04}$), *TalckII β* ($q = 0.0017$), *Talpp1* ($q = 0.023$), *Talshaggy* ($q = 0.021$, *Talebony* ($q = 0.028$), *Talsirt1* ($q = 0.001$), Talsirt7 ($q = 0.037$) and *Taljetlag* ($q = 0.027$) exhibited oscillatory expression profiles by JTK_CYCLE analysis. Interrogation of these data by ANOVA showed *TalCK11β* ($F_{7,30} = 3.284$, $P = 0.014$) *Talpp1* ($F_{7,30} = 3.99$, $P = 0.005$), *Talsirt1* ($F_{7,30} = 4.91$, $P = 0.002$) and *Talslimb* ($F_{7,30} = 3.19$, $P = 0.016$) to be differentially expressed over time. *Talper* mRNA accumulation increased approximately 6-fold during early night and peaked towards the middle of the subjective night (CT18-21) before diminishing at CT24 reaching a nadir at CT6. This profile is approximately antiphasic to *TalckII β*, *Talpp1*, *Talsirt1 and Talslimb* that all showed peaks in expected daytime. *Talcry2* and *Talbmal1* respective contigs also showed antiphasic relationships to *Talper*. However, JTK_CYCLE analysis did not reveal significant rhythmicity in the *Talcry2* transcript.

Given the robust rhythm of *Talper* expression we chose this transcript with which to validate our RNAseq expression data by quantitative PCR. RNA levels of *Talper* in samples taken from an independent time-course experiment show peak expression (∼3-fold change from CT3) at CT19 ($F_{7,32} = 4.54$, $P = 0.01$, see Fig. S27). These data support those revealed by RNAseq with peak expression in the early to mid night albeit with a slightly lower amplitude.

Reads for *Talcwo* were below levels acceptable for analysis and excluded from further investigation.

## DISCUSSION

*Talitrus saltator* exhibits intriguing clock-driven behavioural phenotypes, including circadian locomotor rhythms (*Bregazzi & Naylor, 1972*; *Ugolini et al., 2007*) and time-compensated solar and lunar navigation (*Ugolini et al., 1999*; *Ugolini, Melis & Innocenti, 1999*; *Ugolini, 2003*; *Ugolini et al., 2007*). Their abundance and suitability for behavioural experimentation make them an excellent model for comparative clock analysis but to make the species more genetically tractable we set out to define the neuromolecular components of its circadian clock. Initially, our strategy included homology cloning and RACE PCR to

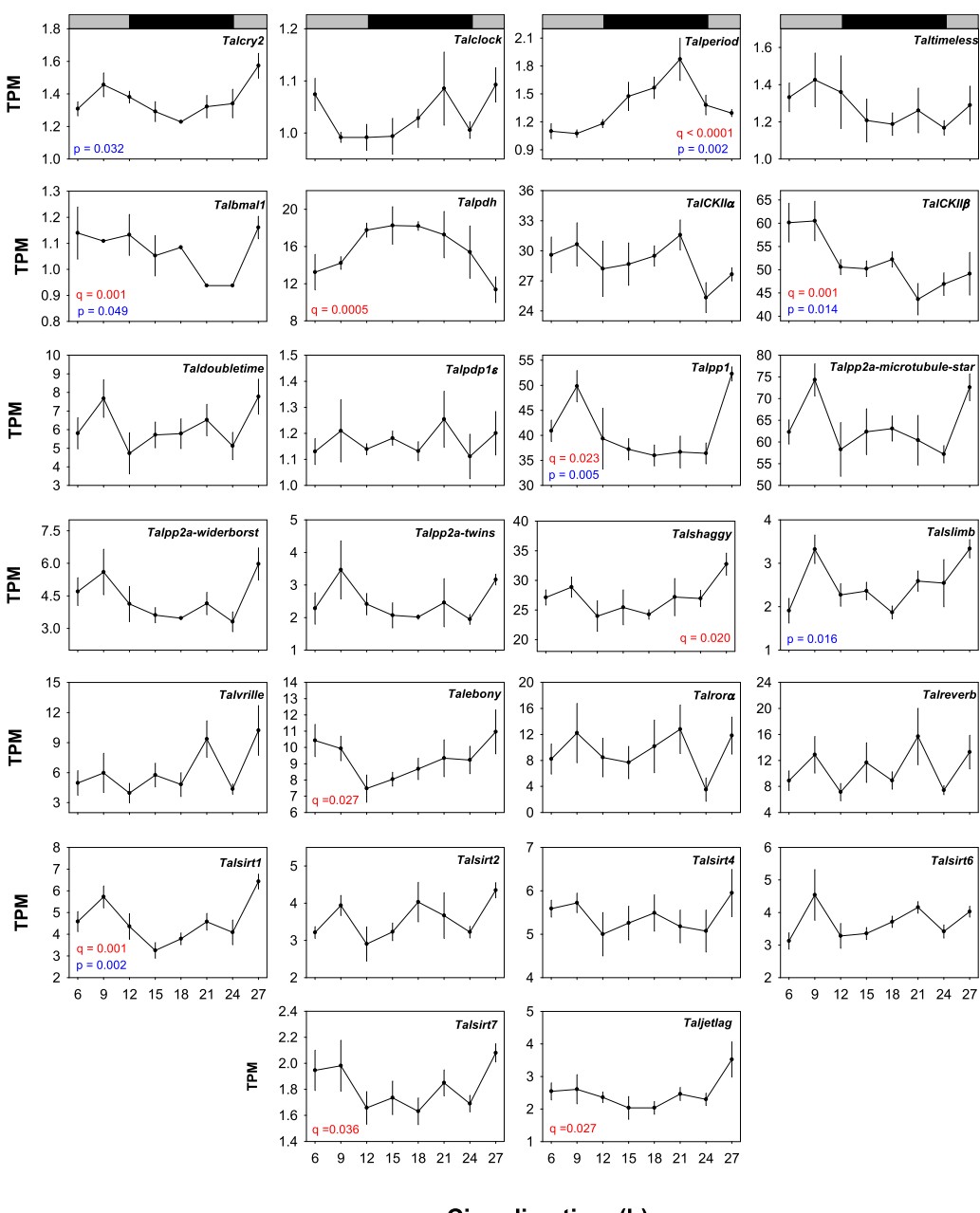

**Figure 4** **Temporal expression profiles of gene transcripts in the brains of free-running *Talitrus salta-tor*.** Plots show TPM values of each identified core clock and clock-associated genes over a 24 h period. Data points represent mean TPM values $+/-$ SEM ($N = 4$ biological replicates, except CT18 where $N = 3$). Significance values are shown in red for JTK_YCLE and in blue for ANOVA. Grey and black bars indicate expected daytime and nighttime, respectively.

sequence full-length *Talcry2*, including the 3′ and 5′ UTRs, but we subsequently exploited RNAseq technologies to sequence the cerebral ganglia transcriptome and identify 'core' clock genes and clock related genes, known in other species to play a role in normal clock functioning.

The quality of the Trinity assembled transcriptome was evaluated in a number of ways. First, we set out to maximise coverage of temporally regulated transcripts by harvesting brain tissue across a complete day-night cycle, and from behaviourally rhythmic animals. The clock gene transcripts identified from our assemblies and the proportion of assembled contigs for that encoded full-length proteins (17 out of 28) suggest we satisfied this aim. Second, standard cloning and sequencing of several genes, including *Talcry2*, *Talper* and *Talbmal1* and *Talpdh-I* resulted in 100% nucleotide alignment with the corresponding contig coding region. Third, confidence in the identity of our contigs as clock gene transcripts was provided by the identification of conserved functional motifs in each conceptual translation. This was reflected also in the BLAST searching and high confidence levels of the search returns (expressed as *e*-values) from the NCBI nucleotide and protein databases as well as FlyBase. Taken together, these attributes indicate that the transcriptome is a faithful representation of all genes expressed in this tissue across a day-night cycle. Furthermore, Blast2GO analysis of the assembled contigs resulted in broadly similar numbers of GO terms and categorisation pattern to previous studies (*Lenz et al., 2014*; *Menon et al., 2012*; *Xie et al., 2012*). The few differences observed between GO term numbers of *T. saltator* to other non-model species can be attributed to transcriptomes being compared across different tissues of diverse species at different stages of development and subjected to various environmental conditions.

Clock genes are well conserved across diverse taxa (*Tauber & Kyriacou, 2005*; *Tauber et al., 2004*) and the discovery of structurally similar transcripts in *T. saltator* was expected. Nevertheless, our data yielded some interesting comparative insights. In *T. saltator* we failed to find a CRY1 but the CRY2 showed very high sequence similarity to that of the isopod *Eurydice pulchra*, also lacking a CRY1 but possessing a mammalian-like CRY2 that has been demonstrated *in vivo* to function as the principle negative repressor BMAL1:CLK transcriptional activity (*Zhang et al., 2013*). Given the close relationship of *T. saltator* to *E. pulchra* it is tempting to speculate that the CRY2 we describe is also a negative regulator in the core oscillator system.

Whilst *Drosophila* do not have a CRY2, other insects, such as the monarch butterfly, *Danaus plexippus* have both CRY1 and CRY2 (*Zhu et al., 2005*). This is considered to be the ancestral state- Cry2 being lost in fruit flies (*Sandrelli et al., 2008*). However, peracarid crustaceans, that presumably predate insects, have only a CRY2, offering alternative evolutionary perspectives on the central clock mechanism. However, the evolutionary status of cryptochrome is likely highly complex; for example, and in contrast to *E. pulchra* and *T. saltator*, the copepod crustaceans, *Calanus finmarchicus* and *Tigriopus californicus* have been shown to express a putative CRY1 transcript (*Lenz et al., 2014*; *Nesbit & Christie, 2014*) whilst some insects such as the flour beetle, *Tribolium castaneum* and the honey bee, *Apis mellifera* have only a mammalian CRY2 (*Rubin et al., 2006*).

Using the *Drosophila* CYC as a search term we elucidated a CYC-like sequence in *T. saltator* containing a bHLH and 2 PAS domains, but differing from *Drosophila* in that it also contained a C-terminal transactivation domain, on the basis of which we assigned the *T. saltator* sequence as a vertebrate-like BMAL1. A very similar structural scenario was described for BMAL1 in *E. pulchra* and the water flea *Daphnia pulex* (*Zhang et al., 2013*).

In *E. pulchra* deletion of the transactivation domain resulted in severely attenuated functionality (of the EpBMAL:EpCLK dimer) in a cell-based *in vitro* luciferase assay (*Zhang et al., 2013*). Again, whilst we have no experimental evidence to support the notion, the close structural and phylogenetic relationship between *T. saltator* and *E. pulchra* hint at a similar role for TalBMAL1.

Although our data reveal most of the clock gene candidates we had anticipated, some such as *Taltim* and *Talclk* were incomplete. However, given the presence of the conserved domains identified within the transcripts, we are confident that these partial sequences are the products of the *Taltim* and *Talclk* genes in *T. saltator*. Manual cloning and sequencing strategies are necessary to reveal the full-length sequences to these. We also uncovered two unusual sequences identified as *Talpdh*. For TalPDH-I the N-terminal end of the mature hormone contains the conserved NSE/ALINSSLLG signature but, the remaining sequence extends beyond the expected 18 residues that define PDHs, including a further 14 residues, and lacking an amidation signal. A second contig discovered with relaxed search stringency also exhibited features of PDH-like peptides with a 23 amino acid signal peptide, a 43 residue PPRP and a 23 amino acid mature PDH with C-terminal amidation. Concentrated efforts to locate other PDH candidates in our transcriptome were unsuccessful. Interestingly, a combined transcriptomic and mass spectrometry approach on the transcriptome/neuropeptidome of the krill *Euphasia crystallorophias* (*Toullec et al., 2013*) revealed three PDH isoforms PDH-L$\beta$1, PDH-L$\beta$2 and PDH-L$\alpha$. In this species PDH-L$\beta$1 expresses the characteristic N-terminal 12 residues but has an extended C-terminus, making the mature, amidated peptide 24 amino acids in length. The PDH-L2$\beta$ is almost identical except for Ser$^2$ being substituted for Ala$^2$, reminiscent of the case in *E. pulchra* (*Wilcockson et al., 2011*). TalPDH-II also expresses the conserved N-terminus with a conserved substitution of Leu$^4$ for Ile$^4$. In accord with the structure of PDH prepropeptides, we observed the signal peptide and PPRP, separated from the mature peptide by a dibasic cleavage site (K-R). These PDH preprohormone-like features and confirmation of the contig sequences by standard and RACE PCR and cloning, together with the unusual PDH isoforms reported in euphausiids, allay our initial fears that our sequence data were mis-assemblies or anomalies. The structural and functional significance of PDHs in *T. saltator* are worthy of further investigation.

Many organisms show cyclic changes in gene expression of core and clock-related genes as a function of the TTFL. A feature of biological clocks is that they free-run in constant conditions i.e., the transcriptional/translational activity of the central oscillator persists with a circadian period in DD. We determined the transcriptional dynamics of the *T. saltator* clock genes in behaviourally rhythmic animals by mapping RNAseq sequencing reads for each time-course sample back to the assembled transcriptome and analysing temporal changes in these data by JTK_CYCLE analysis and ANOVA. Of the canonical clock genes only *Talper* and *Talbmal* were indicated as rhythmically expressed by JTK_CYCLE analysis and ANOVA with *Talcry 2* abundance varying in time (ANOVA). In addition, the clock related genes, *TalckII$\beta$*, *Talpp1*, *Talsirt1, Talsirt7 Talebony, Taljetlag, Talshaggy* and *Talpdh-II and* showed oscillating expression using the same

analytical parameters. *TalCK11β*, *Talpp1*, *Talsirt1* and *Talslimb* abundance differences were also noted over time (ANOVA).

We chose the JTK_CYCLE algorithm because it is reported to be very robust to outliers and returns low numbers of false-positives (*Hughes, Hogenesch & Kornacker, 2010*). In our study we sacrificed an extended sampling period for increased biological replicates and three-hour sampling intervals. We are confident that this approach, coupled with the JTK_CYCLE data treatment yielded genuinely cycling genes that offer suitable targets for further exploration.

We caution that, it would be entirely speculative to draw conclusions on the functional significance of the rhythmicity or phase relationships of cyclic expression identified in the current study. Although circadian genes and proteins are conserved and the central tenet of the TTFL serves to describe the basis of rhythmicity across taxa, simple interspecific comparisons are confounded by the fact that clocks do show nuances in their organisation of the TTFL, e.g., in *E. pulchra* only *Eptim* shows robust cycling in animals expressing circadian and circatidal behaviour and yet *Epper*, which is non-rhythmically expressed, appears essential to circadian phenotypes (*Zhang et al., 2013*). Indeed, rhythmicity in gene expression itself is not a prerequisite for a functional clock (*Lakin-Thomas, 2006*). The tremendous scope for complex interplay of gene products and their cognate proteins will only be revealed by more targeted experimental approaches in non-model species. Nevertheless, our data indicate that sub-sets of clock and clock-related genes in *T. saltator* continue to cycle in free-running conditions and thus represent tantalising targets for investigation to uncover the role of these genes in either circadian or circalunidian (24.8 h) rhythms. For example, *T. saltator* has the capability to orientate by the sun and moon suggesting they may have separate daily and lunidian clocks (*Ugolini et al., 1999*). Indeed a precedence for independent clock mechanisms in marine organisms has been set, e.g., *E. pulchra* has been shown to possess a dual clock system with separate clocks orchestrating circatidal and circadian phenotypes and the marine worm *Platyneries dumerilli* employs separate mechanisms to keep circadian as well as lunar time (*Zantke et al., 2013*). The sequence data and expression analysis described here provide a foot-hold in a behaviourally tractable system from which we can address these questions.

In conclusion, we have sequenced the entire brain transcriptome of an ecologically important beach crustacean that exhibits intriguing clock-controlled phenotypes and report the sequences of key clock and clock-related genes that are likely key players in these phenotypes. Our dataset is one of only very few to describe the putative clock mechanism in a non-model crustacean and contribute to the growing inventory of crustacean and non-model transcriptomes that may have wide-ranging utility in the research community.

## ACKNOWLEDGEMENTS

Thanks to Justin Pachebat, (Aberystwyth University) for his advice and help with initial transcriptome sequencing strategies.

### Funding

This work was supported by a Natural Environment Research Council, UK grant NE/K000594/1 awarded to D.C.W. The funders had no role in study design, data collection and analysis, decision to publish, or preparation of the manuscript.

### Grant Disclosures

The following grant information was disclosed by the authors:
Natural Environment Research Council, UK: NE/K000594/1.

### Competing Interests

The authors declare there are no competing interests.

### Author Contributions

- Joseph F. O'Grady conceived and designed the experiments, performed the experiments, analyzed the data, wrote the paper, prepared figures and/or tables, reviewed drafts of the paper.
- Laura S. Hoelters performed the experiments, analyzed the data, wrote the paper, prepared figures and/or tables, reviewed drafts of the paper.
- Martin T. Swain analyzed the data, reviewed drafts of the paper.
- David C. Wilcockson conceived and designed the experiments, performed the experiments, analyzed the data, contributed reagents/materials/analysis tools, wrote the paper, prepared figures and/or tables, reviewed drafts of the paper.

### DNA Deposition

The following information was supplied regarding the deposition of DNA sequences:
NCBI SRA and GenBank. Assembly project has been deposited at DDBJ/EMBL/GenBank under the accession GDUJ00000000.

### Data Availability

Figshare: https://figshare.com/s/a2513243c63bf557b720.

### Supplemental Information

Supplemental information for this article can be found online at http://dx.doi.org/10.7717/peerj.2555#supplemental-information.

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
