# Peer review of "Identification and temporal expression of putative circadian clock transcripts in the amphipod crustacean Talitrus saltator"

_PeerJ, doi:10.7717/peerj.2555_

## Round 0.1 · original submission · Minor Revisions

Please heed the reviewers' comments, especially Reviewer #2, when submitting a corrected version of the manuscript.

·

Basic reporting

In two places in the introduction there are broken sentences, lines 95-96 and 100-104, that should be corrected.
Words that are spelled correctly but are being misused should be corrected, “understating” should be “understanding” (line 48) and “ally” (line 626) should be “allay”.
The introduction states (lines 115-116) that the clock-controlled phenotypes of T. saltator are “critical for its survival”; while the presence of rhythmic behaviors undoubtedly increases the animal’s fitness, whether they are essential is a different question. Unless there is evidence in the literature that can be cited, this conclusion should be toned down.
Methods would be more complete if the US companies cited also had the city identified, such as Trikinetics and Actimetrics.

Experimental design

no comments

Validity of the findings

Line 559: It may be misleading to say that conserved functional domains were “characterised”. This term might imply to some readers that some functional determination was made to confirm, such as enzymatic or binding activities. It would be less ambiguous to say that that the domains were “identified”.

Reviewer 2 ·

Basic reporting

The manuscript adheres to PeerJ policies. It is written well, except for several long sentences which require rephrasing. In addition I have a few minor grammatical corrections to suggest.
The article conforms with the PeerJ templates except for the abstract which is divided into - Background, Results and discussion. As far as I can see there should also be a methods section.
The figures are good, clear (in the separate files) and relevant.

Line 23-25 - sentence too long. Rephrase.
Line 48 - understanding
Line 73-78 - sentence too long. Rephrase.
Line 95 - leads to
Line 101 - suggesting
Line 100-104 - sentence too long. Rephrase.
Line 106 - remove sandy (appears again next line, same sentence)
Line 120 - and circadian clocks (remove a)
Line 161 - for temporal
Line 563 - all genes
Line 663 - the capability

Experimental design

Overall there is a great deal of work reported, and the methods used are of a high standard.
In the Blast2Go section it is states a sequence identity threshold of 95%. What about sequence coverage? Was a particular cutoff set in order to minimize erroneous annotations? What steps were taken to minimize miss-assembled contigs from being part of the GO analysis? This issue should be clarified.
JTK Cycle was developed by Michael E Hughes in collaboration with Karl Kornacker and John Hosenesch, and not only by the latter authors.
It is unclear how many animals movement contributed to the final results.

Validity of the findings

The behavioural findings are very good and establish the point the authors wish to make. However, as stated above, it is unclear how many animals/observations actually contributed to this. This point should be clarified.
Line 515 - It is stated that JTK_Cycle was limited to a period of 12-27 hours for detecting circadian cyclers. Is this reported correctly or a mistake?

·

Basic reporting

Authors comply with all PeerJ policies and recommendations.

Experimental design

The experimental design is sound an was made complying with international bioethical standards. They use modern techniques to elucidate difficult matters in crustacean molecular biology and succeeded in doing so. The authors use addecuate tools for all the molecular biology and rhythmic analysis and demonstrate the usefulness of this techniques in the specific objective they proposed.

Validity of the findings

The finding are valid, well explained and discussed

Additional comments

In my opinion, the authors showed the importance of studying ecological relevant animal models and in doing so, they used a plausible way to tackle the problem of not having the complete DNA sequence of this animal. I must encourage the authors to use the chronobiology terminology more widely used to allow all readers to understand better their chronobiological findings.

---

## Round 0.2 · accepted · Accept

Your paper is now accepted for publication in PeerJ.

Reviewer 2 ·

Basic reporting

Following the authors corrections and clarifications I have no further issues. I think that the paper is reported well, adheres to all PeerJ policies, and is structured properly.

Experimental design

Same as above. The issues relating to the number of animals used in the behavioural experiments, as well as the annotation parameters, have been adequately clarified.

Validity of the findings

Same as above.

Additional comments

The paper is of much interest and is written well. I fully support the acceptance and publication of this manuscript to PeerJ.

·

Basic reporting

The authors have made all the amendments suggested.

Experimental design

The authors have made all the amendments suggested and discussed some of my comments about them. I agree with their answers to my questions.

Validity of the findings

No comments.

Additional comments

The authors have made all the changes suggested or, in some cases have discussed why they believe is more correct a concept or idea. I agree with them and I think the final draft has been improved and comply with PeerJ standards and policies.